# Understanding preferences for behaviour change support as part of the NHS Health Check: a qualitative study with adults from underserved minoritised ethnic communities

Sophie Griffiths ![ORCID],[1] Yvonne K Bartlett ![ORCID],[1] David P French ![ORCID],[1] Brian McMillan ![ORCID] [2]

¹Centre for Health Psychology, The University of Manchester School of Health Sciences, Manchester, UK
²Centre for Primary Care and Health Services Research, The University of Manchester School of Health Sciences, Manchester, UK

**Correspondence to**
Sophie Griffiths;
sophie.griffiths@manchester.ac.uk

## ABSTRACT

**Introduction** NHS Health Checks (NHSHCs) provide individuals with cardiovascular disease (CVD) risk scores alongside advice and signposting to behaviour change support. A particular problem is that the support people receive is often poorly delivered, absent or not tailored to the needs of people in deprived communities, which risks exacerbating health inequalities. Improving this support is critical if NHSHCs are to achieve their goals of prevention and equity.

**Objectives** To explore needs and preferences for behaviour change support among adults in deprived areas, using a digital prototype presenting CVD risk information and signposting to services.

**Design** A longitudinal qualitative study involving focus groups and semi-structured follow-up interviews.

**Participants and setting** Adults from minoritised ethnic groups eligible for NHSHCs, recruited online and through a community centre, with both methods targeting high-deprivation areas.

**Method** Participants were first shown the digital prototype in focus groups to generate discussion. Follow-up interviews captured more in-depth reflections on needs for behaviour change support. Data were analysed using reflexive thematic analysis.

**Results** We conducted four focus groups and 20 follow-up interviews with 23 adults, predominantly of South Asian ethnicity living in areas of high deprivation. We developed three themes: (1) Trusted information to counter confusion and misinformation; (2) Support that makes change feel possible and meaningful, through culturally and personally relevant advice that addresses unhelpful beliefs about risk reduction and behaviour change and (3) Ensuring access to inclusive, socially connected environments that feel supportive and conducive to action.

**Conclusions** For minoritised ethnic adults in deprived areas, NHSHC support should build on everyday practices and foster positive perceptions of services. Alongside service-level changes, policy action is needed to remove structural barriers (eg, cost, safety) that limit people's ability to act on advice. Such changes could enhance the programme's contribution to reducing inequalities in CVD prevention.

## STRENGTHS AND LIMITATIONS OF THIS STUDY

⇒ This study addresses a key gap in NHS Health Check (NHSHC) literature by recruiting predominantly Asian/Asian British women from deprived communities, a group under-represented despite heightened risk.

⇒ A digital prototype presenting cardiovascular disease risk in a realistic NHSHC format was used to explore support preferences, even though the risk information was hypothetical.

⇒ Focus groups facilitated discussion of participants' support needs, with follow-up interviews providing further reflection and depth.

## INTRODUCTION

Cardiovascular disease (CVD) is the leading cause of death worldwide, responsible for nearly one-third of all annual deaths.[1] In the UK, over 7.6 million people live with CVD.[2] Modifiable conditions such as high cholesterol, high blood pressure and type 2 diabetes increase CVD risk and are strongly influenced by social and economic determinants including deprivation.[3] Effective prevention therefore requires engaging people whose health behaviours are shaped by intersecting social and economic pressures.

The NHS Health Check (NHSHC) invites adults aged 40–74 years without established CVD to assess and manage their risk. Uptake is lowest in communities with the greatest burden: socioeconomically deprived neighbourhoods and minoritised ethnic groups.[4] To improve reach, commissioners have extended delivery beyond general practice into community venues such as pharmacies, mosques and pop-up clinics, which tend to attract higher-risk and underserved groups. These venues facilitate NHSHC attendance

because such community assets mobilise social capital, including informal networks, shared norms and trust.[5]

Attendance alone, however, does not guarantee improved cardiovascular outcomes. Benefits depend on whether people understand and act on the advice received.[6] While policy and research have focused on increasing uptake, less attention has been given to the equally critical step of supporting behaviour change. The generic advice often provided in NHSHCs[7] may be particularly ineffective for those facing combined socioeconomic and ethnic disadvantage. Follow-up support also appears under-used: a pharmacy-based study found low uptake of recommended follow-up, representing a missed opportunity to assist those most in need.[6]

Prior qualitative research in a socioeconomically deprived area in south England concluded that CVD prevention support should be codesigned with local organisations, delivered by trusted community members and centred on peer bonds and enjoyment rather than generic health messages.[8] However, that study explored general preventative preferences rather than needs formed after receiving CVD risk information and did not consider how ethnicity and deprivation intersect to compound disadvantage. To address these gaps, we showed adults from minoritised ethnic and socioeconomically deprived backgrounds a digital prototype presenting hypothetical CVD risk information and support options, similar to the format used in the digital NHSHC pilot,[9] providing a realistic prompt for discussing their support needs.

## Aim

To identify needs and preferences for behaviour change support among adults from socioeconomically deprived and minoritised ethnic backgrounds, using a digital prototype to ground discussions in the context of NHSHCs.

## METHOD
### Study design

This longitudinal qualitative study used focus groups and follow-up one-to-one semi-structured interviews. The study received University of Manchester Research Ethics approval (2024-19916-37529). All participants provided written or electronic consent.

### Participant recruitment

Participants were recruited through in-person and online approaches targeting socioeconomically deprived areas. Deprivation was assessed using the English Index of Multiple Deprivation (IMD), which ranks individual's postcodes in deciles ranging from 1 (most deprived) to 10 (least deprived) based on multiple factors including income and employment. The community centre used for in-person recruitment was in Greater Manchester, in an area ranked in the most deprived IMD decile (1) and primarily served South Asian communities. In-person recruitment was facilitated by community centre staff,

who invited eligible participants from their existing networks. Additional recruitment took place via social media platforms including local community groups and classified ad sites, focusing on high-deprivation areas of Greater Manchester.

Eligibility criteria were the same as those for the NHSHC, namely adults aged 40–74 years old with no previous diagnosis of CVD. Individuals outside this age range or with a prior diagnosis of CVD were therefore excluded. We judged the final sample of participants to have sufficient information power[10] based on the study's defined aim, the quality of dialogue and the alignment of participant characteristics with the research focus on individuals from minoritised ethnic groups living in deprived areas.

No participant was required to interact directly with the digital prototype, which was demonstrated by the researcher via screen share during in-person or virtual sessions. This approach, combined with recruitment through a community centre, reduced potential bias toward participants already comfortable with technology.

### Data collection

Focus groups explored initial responses to the digital prototype, beginning in October 2024 and ending in April 2025. Participants were told that in practice, risk information would be linked to electronic health records. Discussions covered understanding of CVD risk, attitudes towards behaviour change, perceptions of signposting and preferred support types. A short demographic questionnaire was completed at the end of each group.

Participants were invited to a one-to-one follow-up interview at least 1 week after the focus group to reflect on the prototype and discuss personal needs and preferences in greater depth.

All focus groups and interviews were conducted by the first author (SG) who has expertise in qualitative methods and behaviour change. Sessions were held via Zoom or at the community centre, with an interpreter available for Urdu-English translation as needed. The interpreter was briefed on the study aims, confidentiality and the importance of verbatim translation. SG spoke directly to the participant, and the interpreter translated in the third person. The prototype was iterated between each focus group in line with participant feedback.

Participants received a £30 retail voucher for the focus group and an additional £15 voucher for the follow-up interview as a thank you.

### Materials

We used questionnaires (Qualtrics XM or paper) to determine participant eligibility and collect the following demographic information: age (years), gender and ethnicity (2011 England and Wales Census categories), highest education qualification and full postcode.

The digital prototype of an app was developed by a specialist Digital Health Software Team based at the University of Manchester[11] using Marvel software.[12]

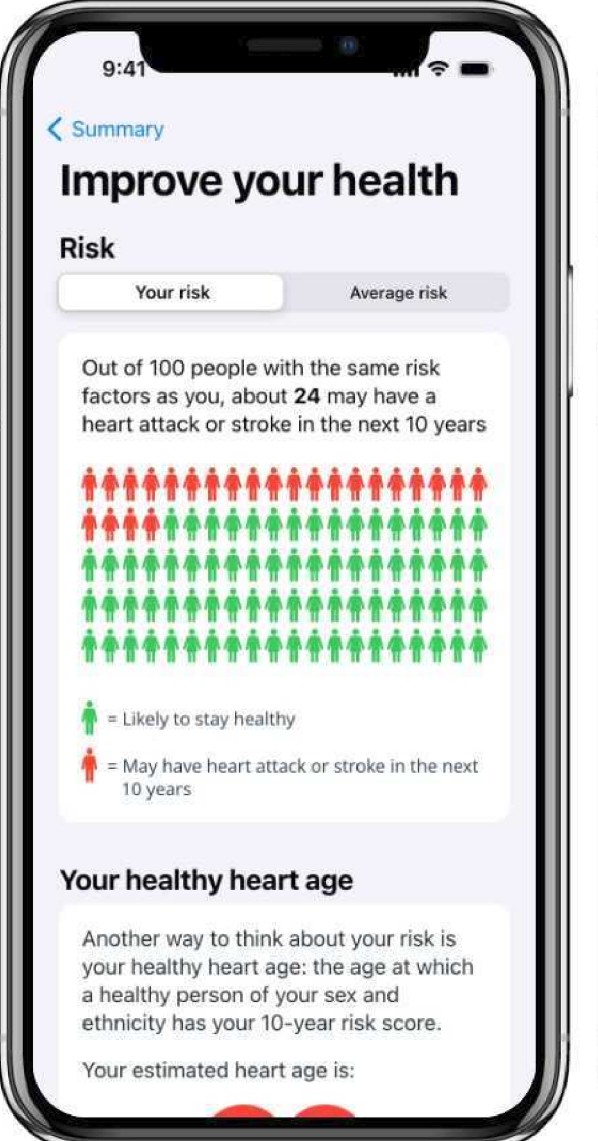
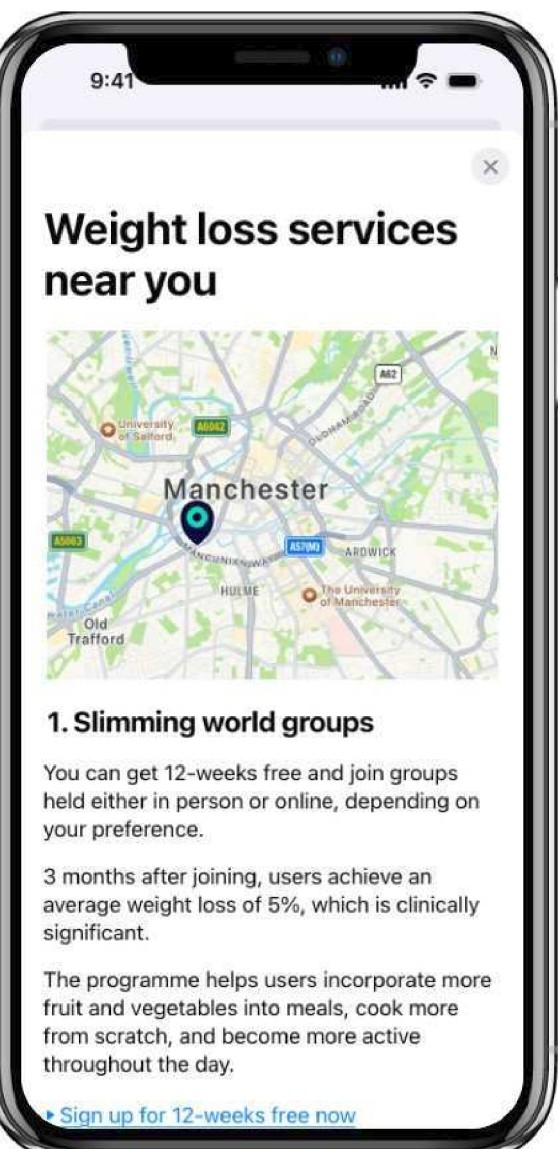

**Figure 1** Screenshots of the digital prototype presenting cardiovascular disease (CVD) risk information and signposting to behaviour change support. The first screen (left) shows the cardiovascular risk result for a hypothetical user. The second screen (right) shows signposting to publicly available local organisations (eg, Slimming World).

The first iteration was informed by patient and public involvement (PPI), literature on risk communication[13–15] and psychological theory.[16 17] Each iteration was refined through a separate codesign process, with details described in a related study. It included visual CVD risk displays alongside a simulated CVD risk estimation tool, illustrating potential risk reductions from health behaviour changes based on a cardiovascular risk prediction algorithm currently under development,[18] as well as brief cardiovascular health education (including national physical activity guidelines) and referral/signposting to weight management services and digital tools (see figure 1 and online supplemental material 1).

A focus group schedule guided participants through the prototype and facilitated discussions specific to its design and preferences for support after receiving CVD information, which were explored in more depth in

interviews. Both focus group and interview topic guides were developed based on previous literature to comprehensively address the research aims and were iteratively refined following feedback from an initial subset of participants to ensure clarity and relevance. The topic guides are provided in online supplemental material 2.

### Patient and public involvement
We involved PPI members from the development stage to help shape the study focus and priorities of this research.[19] All participant-facing documents were reviewed to enhance accessibility. In July 2024, an online session with three PPI members informed the first prototype iteration.

### Analysis
Focus groups and interviews were audio-recorded, transcribed verbatim (including contributions mediated by

the interpreter) and anonymised. Transcripts were analysed using reflexive thematic analysis.[20] Initial coding by SG focused on semantic codes, capturing explicit statements, while later stages of analysis became more interpretative, using latent codes to explore underlying meanings and contextual influences on participants' accounts.

Reflexivity was maintained through reflexive notes, an audit trail documenting analytic decisions and regular discussions of interpretations of the data with coauthors.[20] In line with this reflexive approach, SG considered how her own identity as a white woman may have shaped interactions with participants—most of whom were also women. This shared gender identity may have built rapport and encouraged participants to share more openly. While SG's outsider perspective in relation to cultural background may have limited the full appreciation of cultural nuances, it also encouraged a more open approach to clarifying participants' perspectives and avoiding assumptions. This outsider status may have prompted participants to elaborate more fully, resulting in richer data. Member-checking was not undertaken, as verification-oriented approaches are conceptually inconsistent with reflexive thematic analysis.[21] Credibility was instead ensured through transparent documentation and reflexive, collaborative analytic processes consistent with the epistemological foundations of reflexive thematic analysis.

This manuscript was prepared in line with the Standards for Reporting Qualitative Research guideline[22] and checklist,[23] provided in online supplemental material 3.

## RESULTS

A total of 23 participants attended four focus groups, with 20 completing follow-up interviews. Six were recruited online and 17 via a community centre. Many participants held educational qualifications which did not directly align with UK qualification frameworks (eg, from Pakistan). Given the range of educational systems these data were not categorised further. Sample characteristics are summarised in table 1.

Three themes with subthemes were produced and are outlined in table 2.

### Trusted information as a foundation for engagement

This theme captures participants' need for clear and credible information, linking concerns about widespread misinformation with the desire for personalised advice from trusted sources.

### Navigating health misinformation

Participants reported needing clear, trustworthy health guidance for their health behaviours. Many felt overwhelmed or paralysed by the sheer volume of conflicting advice, particularly when its credibility was uncertain or when they had difficulty securing General Practitioner (GP) appointments to clarify or validate that advice.

**Table 1** Sample characteristics

| Sample characteristic | n=23 |
|---|---|
| Gender | |
| Male | 5 |
| Female | 18 |
| Age (years) | |
| 40–49 | 7 |
| 50–59 | 4 |
| 60–69 | 7 |
| 70–74 | 5 |
| Ethnicity | |
| Asian/Asian British | 22 |
| Arab | 1 |
| IMD decile | |
| 1 (most deprived)–4 | 19 |
| 5–7 | 1 |
| 8–10 (least deprived) | 3 |
| Preferred language* | |
| English | 17 |
| Urdu | 6 |

*An interpreter assisted with translation for participants who preferred to speak Urdu.
IMD, Index of Multiple Deprivation.

*'Let's say you're trying to get healthy. There's so many people, like influencers that give you the wrong information… You feel overwhelmed. You just want to know the truth from someone.'* Nadia, Female, mid 40s.

Information overload was not confined to the internet; contradictory tips exchanged among friends and family also amplified uncertainty.

*'She doesn't follow online; she doesn't trust them at all… From her friends she gets suggestions like, 'Do not eat in the morning', 'Change your flour', 'Do some more walk', 'Do this, do that'—it's sometimes too much, and she's quite*

**Table 2** Summary of the themes and subthemes identified in this study

| Themes | Subthemes |
|---|---|
| Trusted information as a foundation for engagement | Navigating health misinformation |
| | Trust in tailored advice |
| Making change feel manageable and worthwhile | Acknowledgement of everyday efforts and priorities |
| | The toll of guilt, regret and rigid rules |
| | Working with cultural values |
| Creating environments where change feels possible | Safe, inclusive and accessible environments |
| | The motivating power of human connection |

*confused what she needs to do… That's why she wants two-three simple things from an expert that she could follow.* Lubna via interpreter, Female, late 50s.

The lack of accessible, authoritative advice meant navigating multiple, often inconsistent information sources. In response, participants voiced a clear preference for brief, expert-endorsed guidance to bypass contradictory messages and give them a manageable starting point for behaviour change.

### Trust in tailored advice

Participants valued that the information within the prototype carried the authority of a trusted institution and offered recommendations that felt specific to their own circumstances. Having a single, reliable point of reference spared them the effort of searching online and turned uncertainty into a clear, personalised next step to act on.

*'Some people just don't know what to do. Like you said, if you put on there [the app] guidelines, 'Try this, do this, this', you've got something to look at and then do it. Because if there's nothing to guide you… Nobody's going to look on YouTube or Google… But if it's on your app, your own health app, they'll say, 'Oh this is good for me, I'll try this'. Because it's all about you, and nobody else.* Tahira, Female, mid 50s.

Personalised advice mattered not only for its practicality but also for the relief it provided. When participants felt that guidance had been tailored to them, it meant not having to follow generic, one-size-fits-all guidance, which was sometimes experienced as overwhelming and detrimental to mental well-being.

*'My doctor said, like,'Your cholesterol levels are very high' … I've tried to access information, just been given information sheets but no tailored advice or support in how we can manage or monitor better… you're trying to find information yourself, or tailor it to your way… mentally as well, it has an impact because you know the constant worry…'* Adeel, Male, early 40s.

Participants framed trusted, personalised guidance as both more credible and more emotionally manageable, giving them a clear direction in an otherwise confusing information landscape.

### Making change feel manageable and worthwhile

This theme highlights participants' desire for advice relevant to their daily lives, while noting that unhelpful beliefs and cultural context could undermine its actionability if not addressed.

### Acknowledgement of everyday efforts and priorities

Participants described how standard recommendations felt disconnected from their daily lives. They questioned whether informal and incidental forms of movement embedded in their daily routines counted towards meeting the physical activity recommendations.

*'Even carrying the basket up and down the stairs… you don't realise it, how many times you go up and down.* Zara, Female, early 40s.

Recognising these small, everyday efforts could help motivate change and reduce discouragement, particularly when broader life pressures, like mental well-being, disabilities, cost of living or caring responsibilities, shaped what felt possible to attempt.

*'Especially like if they're looking after somebody elderly or they have kids or, you know, it's too far to go to the gym, it's too expensive, there's lots of barriers.'* Sanjay, Male, mid 40s.

Cultural traditions were an added layer of context shaping what felt manageable and sustainable. For example, foods shared during celebrations were often high in fat and sugar, making dietary changes feel incompatible with cultural norms and practices. Participants were open to learning culturally relevant alternatives to make healthy eating feel more realistic without having to abandon traditions.

### The toll of guilt, regret and rigid rules

The prospect of adopting healthier behaviours often carried emotional weight due to participants' beliefs about CVD risk and behaviour change. Some expressed fatalism, the belief that serious illness was inevitable regardless of current efforts. Others reflected on past choices with regret, expressing a sense that the damage to their health had already been done.

*'If only we'd known all this when we were kids… we were eating all junk. But now… we realise, 'If only I had done that'.* Tahira, Female, mid 50s.

Others described an all-or-nothing mindset, where a single lapse, such as eating unhealthy food, was seen as 'cheating'—a moral framing that imposed pressure and reinforced feelings of failure. Unhealthy eating was perceived to negate other efforts, such as exercise, despite cardiovascular benefits beyond weight loss.

*'When I'm in any exercise session… in my head I'm telling myself I have to eat healthy. I can't cheat or anything. There's no point stretching if I'm going to go home and start eating rubbish.* Mahila, Female, mid 60s.

When prompted to discuss goal setting, participants often expressed scepticism. Setting goals was seen as overwhelming, particularly when goals were misaligned with current capabilities.

*'You can't go from zero to 10 by goal setting. Sometimes it has the opposite effect for me… I prefer just doing natural things that feel comfortable for me… it's kind of like Slimming World, it creates a bit of like 'oh, am I a failure?' … Rather than if you just said 'Oh, you did 2000 last week, that's fantastic. Try to do 10 more'.* Nadia, Female, mid 40s.

Participants also described beliefs about diet and activity that made behaviour change feel unnecessarily

difficult. Some of these reflected oversimplified or inaccurate understandings of what needed to be restricted.

*'Chapati, rice — they make you fat.'* Rubina, Female, 74 years.

Such beliefs could lead to cognitively demanding restrictions that were taxing yet of limited benefit. However, more structured approaches, like calorie counting, which might theoretically reduce the need to eliminate foods entirely, were still overwhelming for some participants.

### Working with cultural values

Although the prototype emphasised clinically recommended behaviours (eg, physical activity, smoking cessation), participants raised culturally familiar practices, such as natural remedies. Although often raised tentatively for validation, these practices remained meaningful and motivating, offering a manageable starting point that could build momentum for further positive change.

*'I found out this before what they said in the morning, if you drink warm water with a little bit of lemon in it… it makes your muscles cleaner, bigger, and then, you know, you wash it all down and then makes your muscles clean…. So last week I started doing the warm water with lemon in it, and I feel a lot better… Then you're thinking, oh, you did this in the morning. You don't want to ruin yourself.'* Hafsa, Female, 65 years.

For some, health behaviours were framed as acts of gratitude or spiritual respect.

*'She feels like our body is actually a gift from God, and we need to look after it, and she is being grateful by looking after it.'* Farah via interpreter, Female, late 40s.

Aligning health messaging with these values can make behaviour change feel personally meaningful and rooted in religious and cultural identity. However, participants' sense of agency could be undermined if their efforts were overlooked. Some expressed disappointment or confusion when, despite avoiding smoking or alcohol, they still faced elevated CVD risk.

### Creating environments where change feels possible

This theme captures how engagement depended on support feeling safe, inclusive and accessible, with human connection providing the social encouragement and accountability needed to sustain change.

### Safe, inclusive and accessible environments

Participants described environmental and structural barriers that made behaviour change and accessing support feel out of reach. Concerns around affordability, discomfort and safety varied but had a shared impact: they undermined confidence, increased the emotional cost of participation and made healthy choices feel less viable, and at times even harmful.

While many participants valued outdoor activity for enjoyment and well-being, safety concerns could negate these benefits.

*'Where we live, we suffer from racist abuse a lot… So you're very scared of stepping outside. You can't really switch off from 'Is someone behind me?', 'What is that person going to do?' It's not a stress-free walk - it's a stressful walk, but then it doesn't really do any good to the heart.'* Rashida, Female, early 60s.

Affordability was another key barrier. Some participants joined subsidised programmes but stopped once funding ended.

*'For 3 months [the exercise class] was free. Because they were not giving them any more vouchers, she said they asked for £25 per month. At that time, she couldn't pay, so that's why she stopped.'* Nargis via interpreter, Female, mid 60s.

Expectations around appearance, body image and other attendees also contributed to apprehension. Some participants avoided classes due to perceived dress codes.

*'That was the reason why I was a bit apprehensive [to attend exercise classes], I thought 'Oh gosh, I'm going to have to go and buy a whole wardrobe of gym clothes.' But when we had our first session and I saw everyone in their normal clothes I thought 'OK, that's alright'.'* Rashida, Female, early 60s.

Others were initially worried about standing out as the least fit in the room but felt reassured once they saw a range of abilities among attendees. For some women, embarrassment about their bodies led them to prefer women-only spaces, which were valued for comfort and support.

Digital exclusion also limited access, with online booking systems proving a barrier for those lacking the skills or confidence to use them.

### The motivating power of human connection

Participants stressed that support was more motivating when it involved social connection and human presence. Group settings and professional guidance reduced the burden of change, made healthy behaviours feel normal and energised participants to act.

*'Even a little bit of walking, it helps. But… I think what it is, it's a little bit of motivation. If one person's doing it, then people will copy… Because nobody wants to do it on their own. So there should be like groups.'* Tahira, Female, mid 60s.

Even those favouring digital resources expressed that these often felt too passive without human involvement.

*'There needs to be… some type of moderation, for example, if there's a health coach or someone working with the individual… Rather than just having information laid out. We have so many apps… but there's no one there telling me or pushing me that it's useful… then I feel like the person can lose track.'* Adeel, Male, early 40s.

Cultural familiarity and representation amplified this effect. Professionals with the same ethnic background made advice feel more relevant and safer.

*'It helps more, if it's Asian… the health advice is more better, when you're there there's more attention for you, you're more used to the South Asian people… It's like [a] comfortable environment for you.* Imran, Male, mid 40s.

Some felt that rapport built through digital interactions could motivate them to attend in-person services. However, practical barriers such as work, childcare and transport mean digital alternatives must also be designed to build a sense of human connection to support behaviour change.

## DISCUSSION
### Key findings

Trusted, personalised information underpins engagement with health behaviour change, reflecting the need for clear, credible explanations that address misinformation and feel personally relevant. Behaviour change also needs to feel manageable and worthwhile through support that recognises everyday efforts, reframes unhelpful cognitions, offers realistic adaptations and embraces cultural values. Safe, inclusive, accessible environments supported by human connection are essential, while policy and service-level action are required to address wider structural barriers.

### Comparison with prior work

NHSHC advice is often generic and inconsistently followed up,[7] leaving many, particularly those from socio-economically deprived groups, without practical guidance. Whereas earlier NHSHC attendees reported being able to find the generic advice provided during NHSHCs elsewhere,[24] our findings indicate that the current information landscape is contradictory and confusing. This challenge is intensified for those with limited GP access to clarify guidance, disproportionately affecting South Asian populations since the shift to digital consultations.[25] Support must therefore help people navigate conflicting information by providing trusted, practical guidance that is personalised to their circumstances.

Consistent with prior work showing a preference for small, incremental changes,[24] we found that acknowledging existing behaviours and tailoring recommendations matters. Participants wanted everyday activities, such as housework and walking, recognised as meaningful contributions. Overlooking these, or valued commitments like abstaining from smoking or alcohol, risks discouragement and undermining self-efficacy, a key factor in risk communication and behaviour change.[26]

Fatalism has been highlighted as influential in health behaviour change[27]; our study shows how it can combine with regret, rigid rules and unrealistic targets to create significant cognitive and emotional strain. All-or-nothing mindsets can trigger guilt that discourages further effort, such as avoiding exercise because benefits beyond weight loss are not recognised. This highlights the need for support to counter unhelpful beliefs and reduce the emotional toll of change by reframing it as a process of learning and adaptation rather than strict adherence to rules. Wider social factors, such as racism, further reduced capacity for self-regulation and compounded these challenges.

Earlier studies also noted that some perspectives within minoritised ethnic groups diverge from the biomedical model of health.[5 28] Our findings support this, and identify ways support can leverage cultural beliefs and ensure acceptability. For example, recognising culturally congruent practices while sensitively explaining other risk factors may reduce feelings of unfairness, preserve self-efficacy and harness cultural values as a driver for change.

Calls for greater representation of minoritised ethnic groups in NHSHC research have highlighted the need to understand how ethnicity and deprivation intersect to shape access and engagement.[29] We addressed this by recruiting adults from minoritised ethnic backgrounds in high-deprivation areas to explore support needs and preferences. The relevance of our findings is reinforced by evidence from other health contexts, such as Iqbal *et al*,[30] who report similar barriers, including limited culturally relevant health advice and safe opportunities for physical activity among pregnant British-Pakistani women. These shared barriers indicate systemic issues, pointing towards the need for both system-level change and tailored support.

### Strengths and limitations

A key strength of this study was the recruitment of predominantly South Asian women living in deprived communities, a group rarely represented in research despite their heightened risk[31] and significant influence on household health practices such as cooking.[32] Using a digital prototype emulated the process of receiving CVD risk feedback and support in a format similar to the digital NHSHC pilot.[9] The realistic format helped participants articulate useful support strategies relevant to both face-to-face and digital delivery. Although the prototype was configured only to signpost to weight-loss services, we encouraged participants to consider how the same approach would apply to any behaviour or support need. Nonetheless, this may have limited responses from those with different health priorities. While the use of hypothetical CVD risk information could be argued to have influenced participants' responses, including the amount or intensity of support they felt they needed, this is unlikely to have affected our findings, as our focus was on the qualities of behaviour change support preferred, which are unlikely to depend on a specific risk level. A further limitation is that recruitment from a single area limits transferability and may not reflect the support needs and preferences of other minoritised or socioeconomically deprived communities.

### Implications

This study adds to the evidence that preventative strategies must be codesigned to be trusted, culturally relevant

and embedded in everyday contexts.[8] For NHSHCs, this means moving beyond generic advice to provide guidance that is personally relevant, credible and responsive to the priorities of underserved groups.

The availability of referral pathways remains important,[33 34] but engagement also depends on whether services feel safe, inclusive and practically accessible. Both real concerns (eg, racism, exclusion) and perceived barriers (eg, assumptions about gym clothing) influence willingness to engage. Commissioners and providers therefore need to prioritise creating supportive environments that feel welcoming, so NHSHCs can function more effectively as gateways to follow-up support and improved health outcomes.[35]

Our findings have important implications for the ongoing development of the digital NHSHC. Evaluation of the pilot identified that some users found the digital signposting process, which directs individuals to follow-up support based on their selected health priorities and perceived barriers, too impersonal.[36] We found that incorporating human support into the digital pathway could help build the rapport that some participants felt they would need to engage with follow-up services. Without this opportunity for discussion, the digital pathway depends on users self-reporting their behaviour change priorities and barriers,[36] which limits opportunities to explore or clarify support needs and, in turn, may reduce engagement with appropriate follow-up support. Incorporating an option for human support into the digital NHSHC could help address this limitation and improve the effectiveness of behaviour change signposting.

Future strategies should build on these findings as a foundation for the codesign and evaluation of behaviour change support for minoritised ethnic groups in deprived areas. Priorities include integrating everyday practices and cultural commitments, addressing unhelpful beliefs about CVD prevention and improving perceptions of support services, while ensuring that follow-up options are safe and inclusive. Alongside policy changes to remove structural barriers, such as cost, these approaches are critical to making the NHSHC programme more equitable and effective for underserved groups. Future research could synthesise findings across multiple studies and map support needs to frameworks such as Capability, Opportunity, Motivation—Behaviour[37] to identify the factors influencing engagement with behaviour change support as part of the NHSHC among underserved populations, providing a foundation for designing targeted interventions.

## CONCLUSION

NHSHC behaviour change support for minoritised ethnic adults in deprived areas needs to be trusted, culturally relevant and accessible, with advice grounded in everyday contexts and values. Embedding these principles can improve engagement, strengthen the programme's role as a gateway to follow-up support and reduce inequalities in CVD prevention.

**Acknowledgements** We thank the staff of the community organisation for their assistance with recruitment and translation, and the community members who participated for their invaluable contributions to this research. We also extend our gratitude to the PPI members for their important input. We thank Simon Foster for his contribution to the design and technical development of the digital prototype used in this study.

**Contributors** SG, the guarantor, led the study design and data collection, coded and analysed the data and drafted the manuscript. YKB, DF and BM contributed to the conceptualisation, supervised coding and analysis and reviewed the manuscript. All authors read and approved the final manuscript.

**Funding** This research was funded by the EMIS National User Group (https://emisnug.org.uk/). The algorithm development was funded by the National Institute for Health Research (NIHR) School for Primary Care Research (SPCR) (reference: NIHR SPCR-2021-2026, grant number 648) and Endeavour Health Charitable Trust. The views expressed are those of the authors and not necessarily those of the NIHR, the Department of Health and Social Care, Endeavour Health or the EMIS NUG.

**Competing interests** None declared.

**Patient and public involvement** Patients and/or the public were involved in the design, or conduct, or reporting or dissemination plans of this research. Refer to the Method section for further details.

**Patient consent for publication** Not applicable.

**Ethics approval** This study involves human participants and was approved by University of Manchester Research Ethics approval (2024-19916-37529). Participants gave informed consent to participate in the study before taking part.

**Provenance and peer review** Not commissioned; externally peer reviewed.

**Data availability statement** No data are available. The data from this study are anonymised but contain context-specific narratives that could risk participant identification; to maintain confidentiality, they are not publicly available and cannot be shared.

**ORCID iDs**
Sophie Griffiths https://orcid.org/0009-0002-0638-8168
Yvonne K Bartlett https://orcid.org/0000-0002-5913-3014
David P French https://orcid.org/0000-0002-7663-7804
Brian McMillan https://orcid.org/0000-0002-0683-3877

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
