## [Reviewer comments · BMJ Open]

ARTICLE DETAILS

Title (Provisional)

Understanding Preferences for Behaviour Change Support as part of the NHS Health Check: A Qualitative Study with Adults from Underserved Minoritised Ethnic Communities

Authors

Griffiths, Sophie; Bartlett, Yvonne Kiera; French, David; McMillan, Brian

VERSION 1 - REVIEW

Reviewer	1
Name	Seenan, Chris
Affiliation and Sport	Glasgow Caledonian University, Faculty of Health Sciences
Date	04-Nov-2025
COI	None

This is a valuable and well-written study addressing an important equity gap in NHS Health Check follow-up. The design and analysis are appropriate, the findings are clear, and the reporting is transparent. I suggest minor revisions to clarify sampling and interpreter use, expand on transferability, and strengthen the link between findings and practical implementation. Once addressed, the paper will make a strong and relevant contribution to public health and health-behaviour literature.

Main strengths

- Clear focus on equity and prevention- addresses a real gap in NHS Health Check follow-up.
- Innovative use of a digital prototype to make discussions concrete and meaningful.
- Appropriate longitudinal qualitative design with transparent, reflexive analysis.
- Good representation of a group often under-represented in research.
- Clear writing and thorough reporting (including SRQR checklist).

Main areas for improvement

1. Sample and transferability:

Most participants were South Asian and recruited from one area. Please expand the discussion of how this shapes transferability to other groups and settings.

2. 'Information power' and sample adequacy:

Be more explicit about how you judged when data were sufficient- what signals or criteria were used.

3. Interpreter role and reflexivity:

Clarify how interpreters were involved, briefed, and how their input was handled in analysis.

4. Trustworthiness:

Add a brief note on how you enhanced credibility (e.g. reflexive notes, peer debrief, audit trail) and why member-checking was or wasn't done.

5. Prototype context:

Make clear that the CVD-risk figures shown were hypothetical and note how this might have influenced participants' views.

6. Digital inclusion and bias:

Acknowledge that using a digital prototype may have favoured participants already comfortable with technology.

Minor comments

- Include a short line in the abstract describing participant profile (mainly South Asian; high-deprivation areas).
- Ensure interpreter use and key demographics (e.g. language, education) are clear in participant table.
- Check any statements that might read as frequency counts ("many participants") and soften if needed.

Recommendation

Minor revisions.

This is a valuable and well-executed study that adds new insight into equitable behaviour-change support for NHS Health Checks. With a few clarifications around sampling, interpreter use, and practice implications, it will make a strong contribution to BMJ Open.

Reviewer	2
Name	Riley, V
Affiliation Education	Staffordshire University, School of Life Sciences and
Date	10-Nov-2025

COI

None

Thank you for inviting me to complete a peer-review of article titled “Understanding Preferences for Behaviour Change Support as part of the NHS Health Check: A Qualitative Study with Adults from Underserved Minoritised Ethnic Communities”. The article was enjoyable to read, and the research is needed to improve the effectiveness of NHSHCs for underserved populations. Please see minor comments and revisions outlined below:

1. The literature in the introduction is minimal, likely due to the restricted word limit. If there is an opportunity to do so, please describe some of the evidence in more detail (i.e., alternative venues for improving uptake of underserved populations, difficult in recruiting underserved groups to the NHSHC programme).
2. “Nadia” features more than other in your first two themes – please consider reducing the number of quotes for this participant and reference others that haven’t been quoted where evidence allows.
3. Please add participant gender to quotes.
4. Can you provide more details about participant characteristics presented as a table if possible?
5. You briefly mention the digital NHSHC pilot in your strengths and limitations – given the focus of digital for the NHSHC programme, please comment on the implications of your findings for the digital NHSHC in your implications section.
6. To consider - I wonder how your findings would map to the COM-B to determine the priorities for improving behaviour change support in underserved communities (i.e., What was more salient; capability? opportunity? motivation?). This may help OHID and NHSHC commissioners to direct their attention to the area in most need of improvement.

VERSION 1 - AUTHOR RESPONSE

COMMENTS	AUTHOR’S RESPONSE
Reviewer 1 comments	
This is a valuable and well-written study addressing an important equity gap in NHS Health Check follow-up. The design and analysis are appropriate, the findings are clear, and the reporting is transparent. I suggest minor revisions to clarify sampling and interpreter use, expand on transferability, and strengthen the link between findings and practical implementation. Once addressed, the paper will make a strong and relevant contribution to public health and health-behaviour literature.	Thank you, we appreciate that you believe our paper will make a valuable contribution to the literature on this topic.
Main strengths • Clear focus on equity and prevention- addresses a real gap in NHS Health Check follow-up.	Thank you for your positive comments.

 • Innovative use of a digital prototype to make discussions concrete and meaningful. • Appropriate longitudinal qualitative design with transparent, reflexive analysis. • Good representation of a group often under-represented in research. • Clear writing and thorough reporting (including SRQR checklist). 	
Main areas for improvement 1. Sample and transferability: Most participants were South Asian and recruited from one area. Please expand the discussion of how this shapes transferability to other groups and settings.	The discussion has been expanded to acknowledge that the transferability of findings is limited by the recruitment of a predominantly South Asian sample from one area. A further limitation is that recruitment from a single area limits transferability and may not reflect the support needs and preferences of other minoritised or socioeconomically deprived communities. (p18).
2. ‘Information power’ and sample adequacy: Be more explicit about how you judged when data were sufficient- what signals or criteria were used.	We have clarified in the manuscript the criteria used to judge the adequacy of the sample to ensure sufficient information power to inform the analysis. We judged the final sample of participants to have sufficient information power¹⁰ based on the study’s defined aim, the quality of dialogue, and the alignment of participant characteristics with the research focus on individuals from minoritised ethnic groups living in deprived areas. (p6).
3. Interpreter role and reflexivity: Clarify how interpreters were involved, briefed, and how their input was handled in analysis.	We have added detail to the method regarding the use of interpreters, to make their involvement from data collection through to analysis clear. Sessions were held via Zoom or at the community centre, with an interpreter available for Urdu-English translation as needed. The interpreter was briefed on the study aims, confidentiality, and the importance of verbatim translation. SG spoke directly to the participant, and the interpreter translated in the third person. (p6). Focus groups and interviews were audio-recorded, transcribed verbatim (including contributions mediated by the interpreter), and anonymised. (p8).
4. Trustworthiness: Add a brief note on how you enhanced credibility (e.g. reflexive notes, peer debrief, audit trail) and why member-checking was or wasn’t done.	Reflexive methods, including reflexive notetaking, an audit trail, and discussions with co-authors are explained in the Analysis section. We also explain our rationale for selecting these methods, including why member-checking was not prioritised (p8). Reflexivity was maintained through reflexive notes, an audit trail documenting analytic decisions, and regular discussions of interpretations of the data with co-authors²⁰... Member-checking was not undertaken, as verification-oriented approaches are conceptually inconsistent with reflexive thematic

	analysis²¹. Credibility was instead ensured through transparent documentation and reflexive, collaborative analytic processes consistent with the epistemological foundations of reflexive thematic analysis. (p8).
5. Prototype context: Make clear that the CVD-risk figures shown were hypothetical and note how this might have influenced participants' views.	We have expanded the Discussion section to clarify that the hypothetical CVD-risk figures may have influenced participants' responses, potentially affecting the amount or intensity of support they felt they needed. However, this likely had minimal impact on our analysis, which focused on the type of behaviour change support preferred, something unlikely to depend on the specific numerical risk presented. Therefore, the findings regarding support preferences remain valid despite the use of hypothetical risk figures (p18). Although the prototype was configured only to signpost to weight-loss services, we encouraged participants to consider how the same approach would apply to any behaviour or support need. Nonetheless, this may have limited responses from those with different health priorities. While the use of hypothetical CVD risk information could be argued to have influenced participants' responses, including the amount or intensity of support they felt they needed, this is unlikely to have affected our findings, as our focus was on the qualities of behaviour change support preferred, which are unlikely to depend on a specific risk level. A further limitation is that recruitment from a single area limits transferability and may not reflect the support needs and preferences of other minoritised or socioeconomically deprived communities. (p18).
6. Digital inclusion and bias: Acknowledge that using a digital prototype may have favoured participants already comfortable with technology.	Thank you for this point. We have clarified in the Methods that no participant was required to interact directly with the digital prototype, as it was demonstrated by the researcher via screen share during both in-person and virtual sessions. Recruitment through a community centre further mitigated potential bias toward participants already comfortable with technology. This clarification has been added to the Methods section to address any concerns regarding digital exclusion in participant recruitment. No participant was required to interact directly with the digital prototype, which was demonstrated by the researcher via screen share during in-person or virtual sessions. This approach, combined with recruitment through a community centre, reduced potential bias toward participants already comfortable with technology. (p6).

Minor comments  • Include a short line in the abstract describing participant profile (mainly South Asian; high-deprivation areas). 	These details on participant profile are now included in the abstract. We conducted four focus groups and 20 follow-up interviews with 23 adults, predominantly of South Asian ethnicity living in areas of high deprivation. (p2).
 • Ensure interpreter use and key demographics (e.g. language, education) are clear in participant table. 	We have included language in the participant table, and the asterisk beneath the table details interpreter use for participants who preferred to speak Urdu. Education was not descriptively summarised in the table; the rationale for this is explained in the Results section. Given the range of educational systems these data were not categorised further. (p9).
 • Check any statements that might read as frequency counts (“many participants”) and soften if needed. 	Thank you for this suggestion. Upon review, we have retained the original wording, as we believe all statements appropriately reflect the proportion of participants contributing each view.
Recommendation Minor revisions. This is a valuable and well-executed study that adds new insight into equitable behaviour-change support for NHS Health Checks. With a few clarifications around sampling, interpreter use, and practice implications, it will make a strong contribution to BMJ Open.	Thank you for your thoughtful review. We are pleased that you believe our paper will make a strong contribution to the journal.
Reviewer 2	
Thank you for inviting me to complete a peer-review of article titled “Understanding Preferences for Behaviour Change Support as part of the NHS Health Check: A Qualitative Study with Adults from Underserved Minoritised Ethnic Communities”. The article was enjoyable to read, and the research is needed to improve the effectiveness of NHHSCs for underserved populations. Please see minor comments and revisions outlined below:	Thank you for your review of our paper. We are pleased that you found it enjoyable to read and appreciate its importance.
1. The literature in the introduction is minimal, likely due to the restricted word limit. If there is an opportunity to do so, please describe some of the evidence in more detail (i.e., alternative venues for improving uptake of underserved populations, difficult in recruiting underserved groups to the NHHSC programme).	Thank you. We have added a brief sentence noting that community venues facilitate NHHSC attendance by drawing on social capital. This strengthens the introduction by clarifying why alternative delivery models are effective for underserved groups. While we chose not to cover all recruitment challenges in detail, these venues are known to help overcome barriers that make engaging underserved populations difficult, providing useful context for the support-related gaps our study addresses. To improve reach, commissioners have extended delivery beyond general practice into community

	venues such as pharmacies, mosques, and pop-up clinics, which tend to attract higher-risk and underserved groups. These venues facilitate NHSHC attendance because such community assets mobilise social capital, including informal networks, shared norms, and trust⁵. (p4).
2. “Nadia” features more than other in your first two themes – please consider reducing the number of quotes for this participant and reference others that haven’t been quoted where evidence allows.	We have reduced the number of quotes from Nadia from four to two, bringing it in line with the number of quotes used for other participants, and have added a quote from a participant who was not previously represented. ‘My doctor said, like, ‘Your cholesterol levels are very high’ ... I've tried to access information, just been given information sheets but no tailored advice or support in how we can manage or monitor better... you're trying to find information yourself, or tailor it to your way... mentally as well, it has an impact because you know the constant worry...’ Adeel, Male, early 40s. (p11). ‘Especially like if they're looking after somebody elderly or they have kids or, you know, it's too far to go to the gym, it's too expensive, there's lots of barriers.’ Sanjay, Male, mid 40s. (p12).
3. Please add participant gender to quotes.	The gender of each participant has been added to all quotes.
4. Can you provide more details about participant characteristics presented as a table if possible?	We have added a table to summarise the sample characteristics (Table 1). (p9).
5. You briefly mention the digital NHSHC pilot in your strengths and limitations – given the focus of digital for the NHSHC programme, please comment on the implications of your findings for the digital NHSHC in your implications section.	We agree that this is an important implication, which has now been elaborated on in the Implications section. Our findings have important implications for the ongoing development of the digital NHSHC. Evaluation of the pilot identified that some users found the digital signposting process, which directs individuals to follow-up support based on their selected health priorities and perceived barriers, too impersonal³⁷. We found that incorporating human support into the digital pathway could help build the rapport that some participants felt they would need to engage with follow-up services. Without this opportunity for discussion, the digital pathway depends on users self-reporting their behaviour change priorities and barriers³⁷, which limits opportunities to explore or clarify support needs and, in turn, may reduce engagement with appropriate follow-up support. Incorporating an option for human support into the digital NHSHC could help address this limitation and improve the effectiveness of behaviour change signposting. (p19).

6. To consider - I wonder how your findings would map to the COM-B to determine the priorities for improving behaviour change support in underserved communities (i.e., What was more salient; capability? opportunity? motivation?). This may help OHID and NHSHC commissioners to direct their attention to the area in most need of improvement	We thank the reviewer for this suggestion. While mapping our findings to the COM-B framework could provide useful insights for prioritising behaviour change support, this is beyond the scope of the current study and could not be done justice within the word limit. We have instead highlighted it as an important area for future research, where synthesising findings across multiple studies could allow more robust identification of the extent to which capability, opportunity, and motivation influence engagement with behaviour change support among underserved populations to inform future interventions. Future research could synthesise findings across multiple studies and map support needs to frameworks such as Capability, Opportunity, Motivation -Behaviour (COM-B)³⁸ to identify the factors influencing engagement with behaviour change support as part of the NHSHC among underserved populations, providing a foundation for designing targeted interventions. (p19).
---	---

¹Braun, V, Clarke, V. Toward good practice in thematic analysis: Avoiding common problems and be(com)ing a knowing researcher. *International Journal of Transgender Health*. 2023;24(1):1–6. doi: 10.1080/26895269.2022.2129597.

VERSION 2 - REVIEW

Reviewer **1**

Name **Seenan, Chris**

Affiliation and Sport **Glasgow Caledonian University, Faculty of Health Sciences**

Date **20-Dec-2025**

COI

The authors have responded comprehensively to all reviewer and editor comments, and the revised manuscript is methodologically robust, clearly reported, and policy-relevant. It makes a valuable contribution to understanding equitable behaviour-change support within NHS Health Checks and is suitable for publication.

Reviewer **2**

Name **Riley, V**

Affiliation and Education **Staffordshire University, School of Life Sciences and Education**

Date

15-Dec-2025

COI

I have reviewed the changes made to the revised manuscript and believe they have adequately addressed reviewers comments. I recommend that this is now accepted for publication by BMJ Open